# Genetic Structure of Hereditary Forms of Diabetes Mellitus in Russia

**DOI:** 10.3390/ijms26020740

**Published:** 2025-01-16

**Authors:** Ildar R. Minniakhmetov, Rita I. Khusainova, Dmitry N. Laptev, Bulat I. Yalaev, Yulia S. Karpova, Roman V. Deev, Ramil R. Salakhov, Dmitry D. Panteleev, Kirill V. Smirnov, Galina A. Melnichenko, Marina V. Shestakova, Natalia G. Mokrysheva

**Affiliations:** Endocrinology Research Center, Moscow 117292, Russia; khusainova.rita@endocrincentr.ru (R.I.K.); laptev.dmitry@endocrincentr.ru (D.N.L.); yalaev.bulat@endocrincentr.ru (B.I.Y.); karpova.yuliya@endocrincentr.ru (Y.S.K.); deev.roman@endocrincentr.ru (R.V.D.); salakhov.ramil@endocrincentr.ru (R.R.S.); panteleev.dmitry@endocrincentr.ru (D.D.P.); smirnov.kirill@endocrincentr.ru (K.V.S.); melnichenko.galina@endocrincentr.ru (G.A.M.); shestakova.marina@endocrincentr.ru (M.V.S.); mokrisheva.natalia@endocrincentr.ru (N.G.M.)

**Keywords:** diabetes mellitus, maturity onset diabetes of the young, neonatal diabetes

## Abstract

Analyzing the genetic architecture of hereditary forms of diabetes in different populations is a critical step toward optimizing diagnostic and preventive algorithms. This requires consideration of regional and population-specific characteristics, including the spectrum and frequency of pathogenic variants in targeted genes. As part of this study, we used a custom-designed NGS panel to screen for mutations in 28 genes associated with the pathogenesis of hereditary diabetes mellitus in 506 unrelated patients from Russia. The study identified 180 pathogenic or likely pathogenic variants across 13 genes (*GCK*, *HNF1A*, *HNF1B*, *HNF4A*, *ABCC8*, *INS*, *INSR*, *KCNJ11*, *PAX4*, *PDX1*, *ZFP57*, *BLK*, *WFS1*), representing 46.44% of the analyzed cohort (235 individuals). The glucokinase gene (GCK) had the highest number of identified variants, with 111 variants detected in 161 patients, 20 of which were identified for the first time. In the tissue-specific transcription factor genes *HNF1A*, *HNF4A*, and *HNF1B*, 34 variants were found in 38 patients, including 13 that were previously unreported. Seventeen variants were identified in the *ABCC8* gene, which encodes the ATP-binding cassette transporter 8 of subfamily C, each found in a different patient; four of these were novel discoveries. Nine pathogenic or likely pathogenic variants were identified in the insulin gene (*INS*) and its receptor gene (*INSR*), including four previously unreported variants. Additionally, we identified 10 previously unreported variants in six other genes among 11 patients. Variants in the genes *GCK*, *HNF1A*, *HNF1B*, *HNF4A*, *ABCC8*, *INS*, and *INSR* were the main contributors to the genetic pathogenesis of hereditary diabetes mellitus in the Russian cohort. These findings enhance our understanding of the molecular mechanisms underlying the disease and provide a solid basis for future studies aimed at improving diagnostic accuracy and advancing personalized therapeutic strategies. This knowledge provides a foundation for developing region-specific genetic testing algorithms and personalized therapeutic strategies, which are critical for future initiatives in precision medicine.

## 1. Introduction

Diabetes is a common disease worldwide and a major challenge due to its association with life-threatening complications. The classification of diabetes mellitus (DM) has a long history and is still evolving. Currently, diabetes is classified into four main types: (1) type 1 diabetes mellitus (characterized by autoimmune destruction of the beta cells of the pancreas, usually resulting in complete insulin deficiency; DM1); (2) type 2 diabetes mellitus (characterized by the progressive loss of adequate insulin secretion by the beta cells of the pancreas; DM2); (3) gestational diabetes (diagnosed in the second or third trimester of pregnancy, without a prior diagnosis of overt diabetes) and other types of diabetes resulting from specific causes, such as monogenic diabetes syndromes (neonatal diabetes and maturity-onset diabetes of the young), or exocrine pancreatic diseases (such as cystic fibrosis and pancreatitis); (4) diabetes induced by drugs or chemicals [1]. The primary challenge with most types of diabetes, which precision diabetic medicine seeks to address, is their heterogeneity in etiology, clinical manifestations, and prognosis [2].

Neonatal diabetes mellitus (NDM) and maturity-onset diabetes of the young (MODY) are the two major forms of monogenic diabetes, which usually result from mutations in genes encoding transcription factors, epigenetic regulators, or other proteins involved in pancreatic development and function [3]. MODY is the most common form of hereditary diabetes, characterized by autosomal dominant inheritance. It represents a clinically and genetically diverse group of endocrine disorders, accounting for up to 5% of all diabetes mellitus cases [4]. Currently, 14 subtypes of MODY have been identified, characterized by clinical variability, genetic heterogeneity, and distinct treatment strategies determined by the molecular causes of specific phenotypes [5]. It is estimated that up to 80–90% of MODY cases remain undiagnosed or are misdiagnosed as type 1 (DM1) or type 2 diabetes (DM2) due to their broad clinical spectrum, which often overlaps with the features of DM1 and DM2 [6,7]. Thus, evaluating the pathogenicity of MODY-associated genes is particularly challenging for rare subtypes and those with low penetrance. Accurate diagnosis, prognosis, and treatment of MODY are critical. Further research is required to investigate the genetic landscape, genotype–phenotype correlations, and the role of non-genetic modifiers in pathogenesis within patient cohorts.

Neonatal diabetes mellitus (NDM) is a rare genetic form of diabetes that manifests within the first six months of life, with an incidence of approximately 1 in 350,000 live births worldwide. Clinically, NDM is divided into two forms: transient and permanent. Transient NDM, accounting for about 50% of all cases, is characterized by spontaneous remission after infancy, although relapses can occur later in life. In contrast, permanent NDM persists throughout life.

The causes of NDM encompass hereditary as well as sporadic cases. The most common genetic causes are abnormalities in the 6q24 locus and pathogenic mutations in genes encoding KATP channels, which are critical regulators of insulin secretion. In addition, epigenetic studies have shown that gestational diabetes mellitus (GDM) can significantly influence neonatal gene expression profiles. According to recent epigenome-wide studies, GDM can alter the methylation of genes involved in insulin regulation and pancreatic development and is associated with an increased risk of NDM in offspring [8,9].

Syndromic forms of NDM, such as Rogers syndrome (TRMA), Wolfram syndrome, Alstrom syndrome, and Wolcott–Rallison syndrome, serve as examples of the genetic heterogeneity of the disease. These forms are accompanied by multiple chronic comorbidities resulting from mutations disrupting the function of several physiological systems. A detailed analysis of the genetic landscape of NDM has been provided in studies by Beltran et al. (2020) [10] and Golshan-Tafti et al. (2024) [11], which highlight the molecular diversity underlying this condition.

Advances in the management of NDM have been made possible by the implementation of high-throughput sequencing technologies, which enable accurate genetic characterization. This facilitates targeted therapies, such as the use of sulfonylureas for patients with KATP channelopathies. Additionally, environmental factors during pregnancy, particularly maternal glycemic control, significantly influence gene expression and disease progression in children. Thus, managing NDM requires an integrative approach that combines personalized therapies targeting genetic and epigenetic factors with consideration of environmental influences. These strategies open new opportunities to improve the diagnosis, treatment, and long-term outcomes of patients with NDM.

According to the second International Consensus Report on Precision Diabetes Medicine (2023), a significant gap exists in the genetic testing of monogenic diabetes in populations of non-European origin [12]. Various studies indicate notable differences in the prevalence of MODY variants across ethnic groups [13]. The growing availability of high-throughput sequencing has enabled the identification of new genes associated with monogenic diabetes; however, the strength of evidence linking these genes to the disease varies. To date, over 30 genes have been identified as causes of monogenic diabetes, including NDM and MODY. Mutations in seven of these genes, including *ABCC8*, *GCK*, *HNF1B*, *INS*, *KCNJ11*, *NEUROD1*, and *PDX1*, are implicated in both NDM and MODY. Additionally, some genes are associated with multiple types of diabetes. For instance, mutations in the insulin gene (INS) have been shown to cause hyperinsulinemia, hyperproinsulinemia, neonatal diabetes mellitus, MODY10, and autoantibody-negative type 1 diabetes mellitus (DM1) [14].

Pathogenic variants of the RFX6, *NKX2.2*, *NKX6.1*, *WFS1*, *PCBD1*, *MTOR*, *TBC1D4*, *CACNA1E*, *MNX1*, *AKT2*, *NEUROG3*, *EIF2AK3*, *GLIS3*, *HADH*, and *PTF1A* genes associated with MODY have been identified relatively recently [15]. However, no clear genotype–phenotype correlations have been established, possibly due to incomplete penetrance [16,17,18,19,20]. Strong genetic evidence confirms the etiological role of *HNF1A*, *HNF4A*, and *GCK* genes in the development of MODY. Recent studies of large population samples have not confirmed the role of *BLK*, *KLF11*, and *PAX4* gene variants in the development of monogenic diabetes. These genes were previously described as causal in the pathogenesis of hereditary forms of the disease [21]. The MDEP GCEP expert group (ClinGen Monogenic Diabetes Gene Curation) has excluded these genes from the list of those associated with monogenic diabetes [22].

Thus, the identification of the molecular pathogenesis of hereditary forms of diabetes is far from complete, and further studies on its genetic architecture, taking into account the population-specific characteristics of patient cohorts, are crucial for advancing research. Understanding the molecular and genetic basis of diabetes in the current state of research is critical to the development of new therapeutic approaches aimed at effective disease management and the provision of accurate medical and genetic counseling to affected families. The presented study contributes to the expansion of knowledge in the field of the molecular pathogenesis of hereditary forms of diabetes, provides insights into the genetic landscape of the disease in Russian patients, and offers valuable information on previously unreported variants, which subsequently enriches the database of pathogenic and likely pathogenic variants in target genes associated with the disease.

The purpose of this work is to search for pathogenic variants of targeted genes in patients from Russia with suspected hereditary forms of diabetes.

## 2. Results

In this study, 180 pathogenic or likely pathogenic sequence variants were identified in 13 genes (*GCK*, *HNF1A*, *HNF1B*, *HNF4A*, *ABCC8*, *INS*, *INSR*, *KCNQ1*, *PAX4*, *PDX1*, *ZFP57*, *BLK*, *WFS1*), affecting 46.44% of participants (235 individuals).

The highest number of variants (111) were identified in the glucokinase (*GCK*) gene, a key enzyme in the glycolytic pathway that catalyzes the conversion of glucose to glucose-6-phosphate in the presence of ATP and Mg^2+^ ions, thereby enhancing the sensitivity of β cells to glucose. These variants were found in 161 unrelated patients, 20 of which were reported for the first time (Table 1, Figure 1 and Figure 2).

The spectrum of genetic variants in the *GCK* gene included five splicing variants, ten nonsense mutations, 86 missense substitutions, seven frameshift mutations, and three deletions, one of which was a large deletion.

In exon 2 of the *GCK* gene, five previously undescribed variants in the nucleotide sequence in the heterozygous state were identified in children with a family history of disease. The variant c.88del (HG38, chr7:44153422del, coverage depth 200×) results in the formation of a stop codon, leading to premature termination of translation p.(Leu30Ter), which is likely to cause a loss of function in the affected gene copy. The variant c.136A>T (HG38, chr7:44153373T>A, coverage depth 388×) was identified, resulting in an amino acid substitution p.(Arg46Trp). This variant occurs in the same codon where another pathogenic variant, p.(Arg46Met) (PMID: 36257325), was previously reported. In silico prediction algorithms provide inconsistent results regarding the impact of this variant on the protein structure and function. The variant c.138del (HG38, chr7:44153372del, coverage depth 440×) causes the deletion of a single nucleotide, resulting in a frameshift and the formation of a stop codon, leading to premature termination of translation after 10 codons p.(Arg46SerfsTer10). This variant was identified in a 9-year-old boy from a family with MODY. This variant is highly likely to result in the loss of function of the affected gene copy. The variant c.164T>C (HG38, chr7:44153345A>G, coverage depth 323×) results in an amino acid substitution p.(Val55Ala). It is located in a highly conserved region, and prediction algorithms suggest a pathogenic effect on the protein. The variant c.169A>G (HG38, chr7:44153340T>C, coverage depth 324×) results in an amino acid substitution p.(Met57Val). It is located in a highly conserved region, within the codon where other pathogenic variants, including p.(Met57Arg), p.(Met57Ile), and p.(Met57Thr), have previously been described. Prediction algorithms consistently indicate a pathogenic effect of this variant on protein structure and function.

In exon 4, a novel variant c.461T>G (HG38, chr7:44150978A>C, coverage depth 156×) was identified in a heterozygous state, resulting in the amino acid substitution p.(Val154Gly). The variant is located in a moderately conserved position, and in silico analysis predicts a pathogenic effect on the protein.

In exon 7, five previously undescribed variants were identified in a heterozygous state, with predicted pathogenic or likely pathogenic significance. The variant c.695C>G (HG38, chr7: 44147818G>C, coverage depth 230×), resulting in the amino acid substitution p.(Ala232Gly), is located in a conserved region, specifically at a codon where another pathogenic variant, p.(Ala232Asp), has previously been reported. The variant c.725A>G (HG38, chr7: 44147788T>C, coverage depth 344×), resulting in the amino acid substitution p.(Glu242Gly), is located in a conserved region, and computational algorithms predict a likely pathogenic impact on the protein. The variant c.801_803del (HG38, chr7: 44147712-44147714del, coverage depth 441×) results in the deletion of three nucleotides, preserving the reading frame, in a non-repetitive region of the protein, leading to the synthesis of a protein with an altered length. The variant c.863T>A (HG38, chr7: 44147650A>T, coverage depth 113×), resulting in the amino acid substitution p.(Leu288Gln), is absent in the gnomAD population frequency database, is located in a conserved region, and is predicted by computational algorithms to have a pathogenic effect on the protein. A previously undescribed variant, c.864-1G>T (HG38, chr7: 44146619C>A, coverage depth 205×), was identified at the acceptor splice site of exon 7. The variant occurs at the same codon as another pathogenic variant, c.864-1G>A, previously reported (PMID: 10753050, 25525159, 28726111). It is highly likely to cause a loss of function of the corresponding gene copy. The SpliceAI algorithm, which predicts the impact of nucleotide variants on splicing site function, classifies this variant as pathogenic.

Two previously undescribed variants in the heterozygous state were identified in exon 8: c.899A>T (HG38, chr7: 44146583T>A, coverage depth 461×), resulting in the substitution of glutamic acid with valine (p.Glu300Val), and the variant c.962C>T (HG38, chr7: 44146520G>A, coverage depth 437×), resulting in the substitution of serine with phenylalanine (p.Ser321Phe). Both variants are located in evolutionarily conserved positions, and computational tools predict their pathogenic effect on the protein. Additionally, the variant c.1020-2A>G (HG38, chr7: 44145732T>C, coverage depth 215×), previously undescribed in the literature, was identified in the splice acceptor site of exon 8, likely resulting in the disruption of the canonical splice acceptor site and the functional inactivation of the corresponding gene copy. SpliceAI and AdaBoost algorithms, designed to predict the impact of nucleotide variants on splicing, classify this variant as pathogenic.

Three novel variants were identified in exon 9 in a heterozygous state, all likely to have a pathogenic effect: c.1117_1118del (HG38, chr7:44145635_44145636del, coverage depth 94×), resulting in a frameshift and a predicted truncated protein p.(Ser373ArgfsTer85), likely leading to the loss of function of the affected gene copy. The c.1190G>C variant (HG38, chr7:44145560C>G, coverage depth 188×) results in the amino acid substitution p.(Arg397Pro). This variant is absent in the gnomAD population frequency database and is located in a conserved codon where other pathogenic variants have been reported. Computational tools predict a deleterious impact on protein function. The c.1225G>A variant (HG38, chr7:44145525C>T, coverage depth 230×) results in the amino acid substitution p.(Asp409Asn). This substitution occurs at a conserved position, and computational analysis predicts a pathogenic effect on the protein.

In exon 10, two additional novel variants were identified: the c.1321T>C variant (HG38, chr7:44145213A>G, coverage depth 209×), results in the amino acid substitution p.(Ser441Pro). This position is conserved and corresponds to a codon previously associated with pathogenic variants, including p.(Ser441Leu) and p.(Ser441Trp). Computational tools predict that this variant will have a deleterious effect on protein function. The c.1327del variant (HG38, chr7:44145207delC, coverage depth 279×) causes a single-nucleotide deletion, resulting in a frameshift and the predicted truncated protein p.(Glu443ArgfsTer171). This frameshift is likely to result in the loss of function of the corresponding gene copy.

A boy with neonatal diabetes and a family history of diabetes was found to have evidence suggesting the presence of a large deletion in a heterozygous state of a segment on chromosome 7 spanning coordinates 44144864-44189228, approximately 44,364 bp in size, encompassing the *GCK* gene (NM_000162.5). Complete deletions of the *GCK* gene have been previously reported as pathogenic in diabetes (PMID:23074679).

We also identified mutations in the family of tissue-specific transcription factor genes, including *HNF1A* (homeobox HNF1A), *HNF4A* (hepatocyte nuclear factor 4 alpha), and *HNF1B* (homeobox HNF1B). Pathogenic variants in these genes are associated with HNF1A-MODY and HNF4A-MODY, subtypes of maturity-onset diabetes of the young, as well as renal cysts and diabetes syndrome (RCAD), respectively.

In the *HNF1A* gene, 27 variants were identified in 32 patients, 9 of which are novel and have not been previously reported. In the *HNF4A* gene, variants were found in 3 patients, 2 of which are novel. Additionally, 4 variants were identified in the *HNF1B* gene in 3 patients, with 2 being novel (Table 2, Figure 3 and Figure 4).

A novel variant, c.197del (HG38, chr12:120978965delA, read depth 146×), was identified in exon 1 of the *HNF1A* gene in a heterozygous state. This variant leads to the deletion of a single nucleotide and causes a frameshift mutation (p.Glu66GlyfsTer89). The variant is located in a conserved position and is likely to result in the loss of function of the corresponding gene copy.

Five previously undescribed variants in the heterozygous state were identified in exon 2 of the *HNF1A* gene, none of which are present in the gnomAD population frequency database. The c.424T>A variant (HG38, chr12:120988930T>A, sequencing depth 238×), resulting in the amino acid substitution p.Ser142Thr, is located in a conserved position. This codon has previously been associated with another pathogenic variant (p.Ser142Phe). The c.433T>A variant (HG38, chr12:120988939T>A, sequencing depth 506×), leading to the amino acid substitution p.Ser145Thr, is located in a codon where another pathogenic variant (p.Ser145Phe) has been previously reported.

The c.438A>T variant (HG38, chr12:120988944A>T, sequencing depth 246×), resulting in the p.Gln146His amino acid substitution, is located in a codon previously associated with other pathogenic variants. The c.481_492del variant (HG38, chr12:120988987_120988998del, sequencing depth 140×), resulting in the deletion of four amino acids without a frameshift (p.Ala161_Thr164del), is located in a non-repetitive region and leads to the synthesis of a protein with altered length. The c.518T>G variant (HG38, chr12:120989024T>G, sequencing depth 248×), resulting in the amino acid substitution p.Val173Gly, is located in a codon where another pathogenic variant (p.Val173Met) has been previously reported. Computational algorithms predict a likely pathogenic impact on the protein for all of the variants described above in exon 2 of the *HNF1A* gene.

A previously undescribed variant, c.1080del (HG38, chr12:120996386del, sequencing depth 409×), was identified in exon 5 of the HNF1A gene in the heterozygous state. This single-nucleotide deletion leads to a frameshift (p.His360GlnfsTer4) and is located in a moderately conserved position, with a high likelihood of resulting in the loss of function of the corresponding gene copy.

A novel variant, c.1207C>T (HG38, chr12:120996640C>T, sequencing depth 355×), was identified in exon 6 of the *HNF1A* gene in the heterozygous state. This variant results in an amino acid substitution (p.Leu403Phe), is located in a conserved position, and is predicted by computational algorithms to have a pathogenic effect on the protein.

A previously undescribed variant, c.1745_1746del (HG38, chr12:120999604_120999605del, sequencing depth 73×), was identified in exon 9 of the *HNF1A* gene in the heterozygous state. This two-nucleotide deletion causes a frameshift (p.His582ProfsTer66) and is highly likely to result in the loss of function of the corresponding gene copy.

In exon 6 of the *HNF4A* gene, a previously undescribed variant c.629C>A (HG38, chr20:44418471C>A, coverage depth 365×) was identified in a heterozygous state, resulting in an amino acid substitution (p.Ala210Asp). Computational algorithms predict this variant to have a pathogenic or likely pathogenic effect on the protein. Additionally, analysis of sequencing coverage suggests the presence of a large deletion in a heterozygous state within a segment of chromosome 20, spanning approximately chr20:44355530–44356128 (598 bp), and encompassing exon 1 of the *HNF4A* gene. This rearrangement is not reported in population frequency databases. The *HNF4A* gene is highly likely to exhibit haploinsufficiency.

In exon 2 of the *HNF1B* gene, a previously undescribed variant c.475C>A (HG38, chr17:37739509G>T, coverage depth 277×) was identified in a heterozygous state, resulting in an amino acid substitution p.(Pro159Thr). This variant is located in a codon where other pathogenic variants have been previously reported. Computational algorithms predict its pathogenic effect on the protein. In the same patient, another variant in this gene was identified: c.491A>G (p.Lys164Arg) (rs1206417995), which is likely pathogenic.

In exon 3 of the *HNF1B* gene, a previously undescribed variant c.599A>T (HG38, chr17:37733767T>A, coverage depth 391×) was identified in a heterozygous state, resulting in an amino acid substitution p.(Asp200Val). This variant is located in a conserved position, and computational algorithms predict its pathogenic effect on the protein.

In the *ABCC8* gene, which encodes the ATP-binding cassette transporter subfamily C member 8, 17 pathogenic or likely pathogenic variants were identified, including four that had not been previously described. Most of these variants are missense mutations (13), with two nonsense and two splicing variants (Table 3).

In exon 12 of the *ABCC8* gene, a previously undescribed variant c.1811T>C (HG38, chr11:17430820A>G, coverage depth 250×) was identified in a heterozygous state, resulting in an amino acid substitution p.(Leu604Pro). This variant is located in a conserved position, and computational algorithms predict its pathogenic effect on the protein. In exon 25 of the ABCC8 gene, another previously undescribed variant c.3110C>T (HG38, chr11:17406940G>A, coverage depth 201×) was identified in a heterozygous state, resulting in an amino acid substitution p.(Thr1037Ile). Computational algorithms predict a pathogenic effect on the protein for this variant.

In a patient with neonatal diabetes, a previously undescribed variant c.3443T>C (HG38, chr11:17404626A>G, coverage depth 499×) in exon 28 of the *ABCC8* gene was identified in a homozygous or hemizygous state, resulting in an amino acid substitution p.(Leu1148Pro). This variant is located in a conserved position and is predicted to have a pathogenic effect.

In a patient with congenital hyperinsulinism, a previously undescribed variant c.4454A>C (HG38, chr11:17394357T>G, coverage depth 256×) in exon 37 of the *ABCC8* gene was identified in a homozygous or hemizygous state, resulting in an amino acid substitution p.(Gln1485Pro). This variant is not reported in the population frequency database gnomAD, and computational algorithms predict it to be pathogenic.

Heterozygous mutations in the *ABCC8* gene have been well-documented in leucine-sensitive neonatal hypoglycemia (OMIM:240800), transient neonatal diabetes mellitus 2 (OMIM:610374), and non-insulin-dependent diabetes mellitus (OMIM:125853), all exhibiting autosomal-dominant inheritance. Heterozygous, homozygous, and compound heterozygous mutations in the *ABCC8* gene have been implicated in familial hyperinsulinemic hypoglycemia (OMIM:256450) and permanent neonatal diabetes mellitus 3 (OMIM:618857), which may present with or without neurological features, and are associated with both autosomal-dominant and autosomal-recessive inheritance patterns. The protein encoded by the *ABCC8* gene is a member of the ATP-binding cassette (ABC) transporter superfamily, serving as a modulator of ATP-sensitive potassium channels and insulin secretion. ABC transporters facilitate the transport of diverse molecules across both extracellular and intracellular membranes. Mutations in the ABCC8 gene are associated with dysregulated glucose metabolism, resulting in the closure of ATP-sensitive channels. This triggers membrane depolarization, subsequent opening of voltage-gated Ca^2+^ channels, and ultimately culminates in insulin secretion. Furthermore, *ABCC8* gene variants can cause hyperinsulinemic hypoglycemia due to inactivating mutations that impair the magnesium-ADP-mediated opening of the channel [23].

We identified nine pathogenic or likely pathogenic variants in the insulin (*INS*) gene and its receptor (*INSR*), including four novel variants not previously reported in the scientific literature. Among these, four missense variants were detected in the INS gene, one of which was identified for the first time (Table 4).

In a patient suspected of MODY, a novel heterozygous variant (HG38, chr11:2160943A>T, c.29T>A, read depth: 143×) in the second exon of the *INS* gene was identified, resulting in the amino acid substitution p.(Leu10Gln). This variant is absent from the gnomAD population frequency database, located in a conserved region, and is predicted to have a likely pathogenic effect on the protein. Heterozygous variants in the INS gene have been reported in insulin-dependent diabetes mellitus 2 (OMIM:125852), hyperproinsulinemia (OMIM:616214), and maturity-onset diabetes of the young type 10 (MODY 10) (OMIM:613370), all exhibiting autosomal-dominant inheritance.

Five likely pathogenic variants in the insulin receptor gene (INSR) were identified in the heterozygous state, three of which are novel and not listed in the gnomAD population frequency database. Heterozygous variants in the *INSR* gene have been associated with insulin-resistant diabetes mellitus accompanied by acanthosis nigricans (OMIM: 610549) and familial hyperinsulinemic hypoglycemia type 5 (OMIM: 609968), both inherited in an autosomal dominant manner.

In a patient with a family history of MODY diabetes, a novel variant, c.322G>T (HG38, chr19: 7267675C>A; coverage depth: 319×), was identified in exon 2 of the *INSR* gene, resulting in an amino acid substitution (p.Val108Phe). This variant is located in a highly conserved position and is predicted to have a pathogenic effect on the protein.

In a child with suspected neonatal diabetes, a novel variant, c.1622A>G (HG38, chr19: 7166393T>C; coverage depth: 368×), was identified in exon 8 of the *INSR* gene. This variant results in an amino acid substitution (p.Asn541Ser) and is computationally predicted to have a pathogenic effect on the protein.

A novel variant, c.3659+1G>C (HG38, chr19: 7120619C>G; coverage depth: 455×), was identified at the donor splice site of exon 20 of the *INSR* gene in a patient with suspected MODY diabetes. This variant likely disrupts the canonical donor splice site, leading to a loss of function of the corresponding gene copy. Predictive algorithms, including SpliceAI and AdaBoost, classify this variant as pathogenic.

In addition, we identified 10 pathogenic or likely pathogenic variants across six genes in 11 patients (Table 5).

Three unrelated patients with neonatal diabetes carried two previously described pathogenic missense variants in the *KCNJ11* gene, which encodes a protein essential for ATP-sensitive K+ channel function in pancreatic beta cells. These channels play a key role in insulin secretion. Additionally, a likely pathogenic variant was identified in the same gene in a patient with suspected MODY diabetes. All variants were found in the heterozygous state. Heterozygous mutations in the *KCNJ11* gene have been associated with type 2 diabetes mellitus (OMIM: 125853), transient neonatal diabetes mellitus type 3 (OMIM: 610582), permanent neonatal diabetes mellitus type 2 with or without neurological features (OMIM: 618856), and maturity-onset diabetes of the young type 13 (OMIM: 616329), all with autosomal dominant inheritance.

In the PAX4 gene, which encodes a transcription factor involved in pancreatic cell differentiation, two previously reported variants with likely pathogenic effects were identified. A missense variant, p.Thr213Met, was found in a patient with an unspecified form of diabetes, and a splicing variant, c.771+3A>G, was detected in a patient with MODY. Computational predictions indicate likely pathogenic and pathogenic effects on the protein, respectively. Heterozygous mutations in the PAX4 gene have been reported in MODY type 9 (OMIM: 612225), type 2 diabetes mellitus (OMIM: 125853) with autosomal dominant inheritance, and in diabetes with a predisposition to ketosis (OMIM: 612227), with both autosomal dominant and autosomal recessive inheritance patterns.

In the *PDX1* gene, responsible for regulating genes involved in pancreatic development and beta-cell function, two novel missense variants were identified. The first, c.417C>G (HG38, chr13: 27924266C>G; coverage depth: 469×), located in exon 2, introduces a stop codon, leading to premature termination of translation (p.Tyr139Ter), with a high likelihood of loss of function for the corresponding gene copy. The second variant, c.533A>C (HG38, chr13: 27924382A>C; coverage depth: 186×), also located in exon 2, results in an amino acid substitution (p.Glu178Ala) at a conserved position, within a codon previously associated with other pathogenic variants. Neither variant is present in the gnomAD database. Computational predictions suggest a pathogenic effect on the protein. Heterozygous mutations in the *PDX1* gene have been reported in susceptibility to type 2 diabetes mellitus (OMIM: 125853) and MODY type IV (OMIM: 606392), both with autosomal dominant inheritance.

In exon 2 of the *ZFP57* gene, a zinc finger protein-coding gene, a heterozygous c.115del variant (HG38, chr6:29676889del, rs758356804, depth of coverage 112×), was identified. This variant results in a single-nucleotide deletion, leading to the formation of a stop codon and premature termination of translation (p.Val39Ter). It is predicted to result in a loss of function of the corresponding gene copy. This variant is listed in the gnomAD v4.0.0 population database at an extremely low heterozygous frequency of 0.00041% (6 heterozygotes). Heterozygous, homozygous, and compound heterozygous mutations in the *ZFP57* gene have been reported in connection with transient neonatal diabetes mellitus type 1 (OMIM: 601410), exhibiting both autosomal dominant and autosomal recessive inheritance patterns. In our case, this alteration was detected in a patient with suspected MODY and a positive family history.

A previously unreported heterozygous variant, c.590C>A (HG38, chr8:11554860C>A, rs150630845, depth of coverage 656×), was identified in exon 7 of the *BLK* gene. The *BLK* gene encodes a non-receptor tyrosine kinase from the Src proto-oncogene family, which is involved in cell proliferation and differentiation. This variant results in the formation of a stop codon and premature termination of translation (p.Ser197Ter). It is predicted to result in a loss of function of the corresponding gene copy. This variant is reported in the gnomAD v4.0.0 population database at a heterozygous frequency of 0.000205% (3 heterozygotes). Heterozygous mutations in the BLK gene have been described in association with MODY type 11 (OMIM: 613375) with autosomal dominant inheritance patterns.

A heterozygous variant, c.1214A>G (HG38, chr4:6301009A>G, depth of coverage 103×), in exon 8 of the WFS1 gene was identified in a father and his child, both of whom were diagnosed with confirmed MODY type 2. This variant results in an amino acid substitution (p.Tyr405Cys) (rs769514478). The variant is situated in a highly conserved region, and computational algorithms strongly predict its pathogenic impact on the protein. It is reported in the gnomAD v2.1.1 population database at a heterozygous frequency of 0.000398% and is annotated as a variant of uncertain clinical significance in the ClinVar database. Heterozygous mutations in the WFS1 gene have been described in association with non-insulin-dependent diabetes mellitus (OMIM: 125853) and Wolfram-like syndrome (OMIM: 614296) with autosomal dominant inheritance patterns.

No additional significant variants matching the search criteria were detected during the analysis.

Our findings contribute to the understanding of the molecular pathogenesis of hereditary forms of diabetes mellitus. The significant number of previously unreported variants in target genes among patients from Russia highlights the need for further comprehensive studies to elucidate their role in disease development.

## 3. Discussion

The wide spectrum of mutations in genes associated with monogenic forms of diabetes results in diverse clinical manifestations, requiring tailored treatment strategies that range from dietary interventions and lifestyle modifications to insulin therapy. A major challenge in most forms of diabetes is their heterogeneity in etiology, clinical presentation, and prognosis, which represents the central aim of precision medicine in diabetes care [2]. Advancements in molecular genetic testing have broadened the classification of diabetes, which was traditionally limited to two primary types, to encompass various hereditary, polygenic, and transitional forms, thereby necessitating updates to the guidelines for differential diagnosis and treatment. The frequency and spectrum of gene mutations associated with hereditary forms of diabetes have been extensively investigated in North America and Europe, with a particular focus on the United Kingdom and Norway. However, large-scale genetic studies remain limited for populations and ethnic subgroups in countries outside of these regions, including Russia [24].

At the Endocrinology Research Center, studies on hereditary forms of diabetes have been conducted using NGS technology over the past decade, with several interim findings already published. In a study of 312 patients (162 boys and 150 girls) aged 3 months to 25 years with suspected MODY, 99 mutations in the *GCK* gene were identified among 129 probands and 77 relatives, 20 mutations in the *HNF1A* gene in 19 probands and 14 relatives, and 8 mutations in the *HNF4A* gene in 9 probands and 3 relatives. Furthermore, the clinical and genetic characteristics of specific MODY subtypes have been comprehensively characterized [25].

Our findings on the prevalence and spectrum of MODY-associated mutations in the Russian population align with global trends, while also highlighting regional genetic characteristics. A comprehensive analysis of MODY in sub-Saharan Africa revealed significant heterogeneity in MODY-related gene variants, with novel mutations identified in *GCK*, *HNF1A*, and *ABCC8* genes. Interestingly, the frequency of *GCK* mutations in African cohorts was considerably lower compared to European and Russian populations, suggesting potential regional genetic influences on MODY subtype distribution. These findings underscore the importance of exploring population-specific genetic landscapes to improve diagnostic accuracy in diverse ethnic groups [26].

In 2019, next-generation sequencing (NGS) was applied to 60 unrelated Russian patients with diabetes and impaired glucose tolerance from Northwest Russia. The analysis included whole-exome sequencing and a targeted panel of 35 genes. Genetic variants were detected in 55% of these patients, with a total of 38 variants identified across the following genes: *GCK*, *HNF1A*, *PAX4*, *ABCC8*, *KCNJ11*, *BLK*, *GATA6*, *WFS1*, *EIF2AK3*, and *SLC19A2*. Notably, 15 of these variants were novel [27].

As a result of a genetic study of patients with suspected MODY in Tunisia, a pathogenic variant in the GCK gene was identified in only one patient out of 17 cases studied (5.9%). Our study revealed a broader spectrum of variants and implicated genes, which may be attributed to the larger size of the patient cohort. A separate study analyzing a panel of 14 MODY-associated genes in 178 patients with a MODY phenotype from Western Siberia identified mutations in *HNF4A*, *GCK*, *HNF1A*, and *ABCC8* genes in 38 patients. Conversely, no pathogenic variants were detected in the *HNF4A*, *PDX1*, *NEUROD1*, *KLF11*, *CEL*, *PAX4*, *INS*, *BLK*, *KCNJ11*, or *APPL1* genes [28].

Studies conducted in Korean populations demonstrated a high prevalence of *HNF1A* and *HNF4A* mutations in individuals with early-onset diabetes, mirroring our findings in the Russian cohort. However, unique mutations were identified in mitochondrial DNA, which are rare in European and Russian populations. This observation highlights the potential for distinct genetic contributions in Asian populations and the necessity of including mitochondrial gene analysis in diagnostic protocols for MODY [29]. Another notable comparison can be drawn with Middle Eastern populations, where the frequency of mutations in the *INS* and *HNF1B* genes is higher compared to other regions. While these mutations are rare in our cohort, the novel variants identified in the Russian population underscore the distinct genetic environment influenced by ethnic and geographical factors. These disparities highlight the necessity for tailored genetic studies to address region-specific clinical needs [30].

Eight unique pathogenic or likely pathogenic variants were identified in eight Greek patients with MODY, resulting in a diagnostic yield of 16%. Five previously unreported pathogenic or likely pathogenic variants were identified in the *GCK*, *HNF4A*, and *ABCC8* genes. Two heterozygous deletions of the entire *HNF1B* gene, confirmed to be de novo, were identified in two patients, increasing the diagnostic yield to 20% [31].

In a study of 42 Chinese patients with suspected MODY, 26 pathogenic or likely pathogenic variants were identified in 25 patients (59.5%). Among these, 15 patients (60.0%) had *GCK* mutations, 4 (16.0%) had *PAX4* mutations, 3 (12.0%) had *HNF4A* mutations, and 1 patient each (4.0%) had mutations in the *INS*, *NEUROD1*, or *HNF1A* genes. Nine mutations (36.0%) were previously unreported [32].

An analysis of a cohort of 160 individuals from the United States with suspected monogenic diabetes identified heterozygous likely pathogenic or pathogenic variants in 37.5% of cases. Among these, 90% were found in either the *GCK* or *HNF1A* genes. Mutations were also detected, though at lower frequencies, in the *PDX*, *HNF4A*, *HNF1B*, and *KCNJ11* genes [33].

A systematic review of MODY mutations worldwide reported that *GCK* and *HNF1A* mutations remain the most prevalent across all populations studied, comprising up to 80% of cases. However, the same review emphasized the underrepresentation of data from Eastern Europe, including Russia, which our study helps to address. By identifying novel mutations in MODY-related genes within the Russian cohort, we contribute to filling this gap and expanding the global understanding of MODY’s genetic spectrum [34].

Thus, the frequencies and spectrum of pathogenic variants and associated genes in hereditary forms of diabetes vary across different regions of Russia. Our study, based on a geographically diverse cohort of patients from various regions, provides a comprehensive snapshot of the molecular architecture of hereditary diabetes in Russia.

Our findings align with those of a study of 224 MODY patients in Turkey, where mutations were identified in the following proportions: 65% in the *GCK* gene, 19% in *HNF1A*, 3.6% in *HNF4A*, 3.6% in *KLF11*, and 3.1% in *HNF1B*. Other mutations were found in the *PDX*, *NEUROD1*, *CEL*, *INS*, *ABCC8*, and *KCNJ11* genes [35]. However, our study identified a significant number of previously unreported variants in these genes.

Recent reviews on MODY genetics have noted the significant overlap in clinical presentations between MODY subtypes and other forms of diabetes, complicating diagnosis. Our study’s use of next-generation sequencing not only validates previously reported global findings but also introduces novel insights into genotype-phenotype correlations specific to the Russian population. These findings emphasize the role of high-throughput sequencing in uncovering unique genetic variants that contribute to diabetes pathogenesis in underrepresented populations.

The identification of pathogenic variants in genes such as *GCK*, *HNF1A*, and *ABCC8* highlights their significance in MODY’s molecular etiology within the Russian population. These findings provide a valuable basis for refining diagnostic algorithms, facilitating earlier detection, and advancing personalized treatment strategies. Moreover, the novel variants identified in this study enhance the understanding of MODY’s genetic landscape on a global scale and serve as a foundation for further studies on genotype–phenotype correlations in underrepresented populations.

In conclusion, our study has expanded the understanding of the molecular genetic basis of hereditary diabetes in Russia. Based on these findings, genetic diagnostic algorithms for the disease are being refined. Our results contribute to the expansion of the database of causative variants for hereditary forms of diabetes, laying a strong foundation for future functional studies.

The limitations of this study include the lack of proven pathogenic significance for some variants and the limited availability of information. Additional limitations include the absence of clinical–genetic correlation analysis, largely due to the high diversity of nucleotide sequence variations and the limited detail of information about the ethnic origin of patients.

## 4. Materials and Methods

The study involved 506 unrelated patients (median age: 13.5 ± 10.15 years) treated at the Endocrinology Research Center, including 240 males (median age: 13 years) and 266 females (median age: 14 years). These patients were suspected of having hereditary forms of diabetes without signs of an autoimmune process, as indicated by the absence of detectable levels of IAA, ICA, GADA, and Zn8 antibodies at disease onset. The inclusion criteria were preserved C-peptide secretion, low demand for exogenous insulin within five years of disease onset, and the absence of severe obesity. Additional cohort characteristics, categorized by the referring physician’s presumptive diagnosis, are presented in Table 6.

Data collection followed an established protocol for this patient category, covering detailed medical histories (including hereditary burden, pregnancy, and delivery history) and measurements of anthropometric and laboratory parameters. Laboratory tests included venous plasma glucose (measured in isolation or during an oral glucose tolerance test [OGTT]), glycated hemoglobin (HbA1c), C-peptide, and insulin. Patients meeting the inclusion criteria provided written informed consent and underwent venous blood sampling into K3/2EDTA tubes. Informed consent was obtained from adult patients or a parent/legal guardian for pediatric participants.

Ethical approval for this study was obtained from the Institutional Review Board (IRB) of Endocrinology research center under protocol number 16 dated 13 September 2023. Written informed consent was obtained from all participants or their legal guardians in accordance with the Declaration of Helsinki. The consent form detailed the purpose of the study, procedures involved, potential risks, and the confidentiality of personal data.

Inclusion criteria: The study population consisted of individuals with various forms of diabetes suspected of having a hereditary origin. Exclusion criteria: Presence of organic pathology of the central nervous system, inclusion of family groups, or identification of relatives through questionnaires and family history data.

Genomic DNA was extracted from peripheral blood lymphocytes using the MagPure Blood DNA kit (Magen, Guangzhou, China). The quantity and purity of the extracted DNA were assessed using a Nanodrop 2000 spectrophotometer (Thermo Fisher Scientific, Waltham, MA, USA) and a Qubit 2.0 fluorometer (Invitrogen, Carlsbad, CA, USA) with the Qubit dsDNA HS Assay Kit.

Sequencing was performed on the Illumina platform using paired-end sequencing (2 × 150 bp). The average coverage depth was 173.4×, with 99.88% of target nucleotides achieving effective coverage >10×.

Whole-genome library preparation (KAPA HyperPlus, Roche, Switzerland) and target DNA enrichment (KAPA HyperCapture, Roche, Switzerland) were performed according to the manufacturer’s protocols. The custom panel targeted the coding regions of 27 genes: *GCG*, *GLUD1*, *WFS1*, *HNF1A*, *GCK*, *INS*, *HNF1B*, *ABCC8*, *HNF4A*, *RFX6*, *PTF1A*, *NEUROD1*, *AKT2*, *ZFP57*, *INSR*, *EIF2AK3*, *PPARG*, *PAX4*, *PDX1*, *GLIS3*, *KCNJ11*, *SLC16A1*, *FOXP3*, *BLK*, *CEL*, *KLF11*, *SCHAD*, and *GCGR*, which have been extensively described in the scientific literature and documented in the OMIM database as being implicated in monogenic forms of diabetes mellitus (DM).

The study utilized next-generation sequencing (NGS) on the Illumina Novaseq 6000 platform (Illumina, San Diego, CA, USA) with paired-end reads (2 × 100 bp). NGS data processing was performed using a standardized automated pipeline, including alignment of reads to the human genome reference sequence (GRCh38), post-alignment processing, variant identification, and quality filtering. Detected variants were annotated for all known gene transcripts using the RefSeq database and pathogenicity prediction algorithms in accordance with the recommendations of the American College of Medical Genetics and Genomics.

Machine learning-based tools, SpliceAI and AdaBoost, were employed to predict the impact of changes in splice sites and adjacent intronic regions. The clinical significance of identified variants was assessed using OMIM and HGMD databases. Their impact on protein structure and function was evaluated with computational tools, including Annovar (http://www.openbioinformatics.org/annovar/ (accessed September 2024)), SIFT (https://sift.bii.a-star.edu.sg/, accessed September 2024), MutationTaster MutationTaster (online version (https://www.mutationtaster.org/, accessed September 2024), and MutPred (version 2.0 (http://mutpred.mutdb.org/, accessed September 2024)), as well as population databases, such as 1000 Genomes, the Exome Aggregation Consortium, and dbSNP [36].

## Figures and Tables

**Figure 1 ijms-26-00740-f001:**
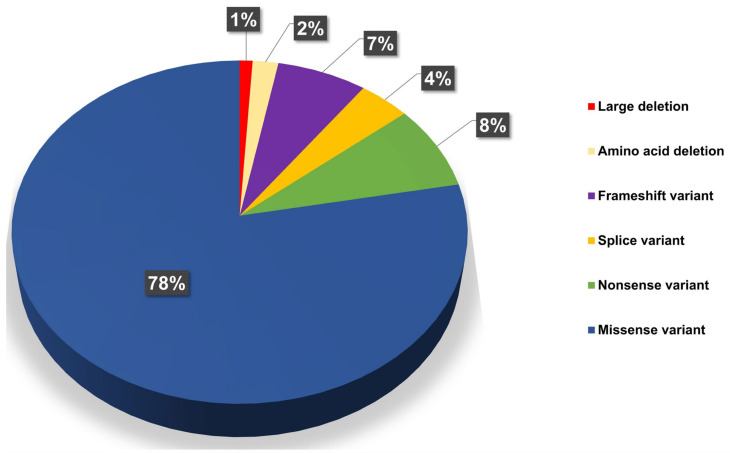
The percentage distribution of different types of genetic variants in the *GCK* gene in the patient cohort. Missense mutations are the most common, while a large deletion is a single variant identified in this gene.

**Figure 2 ijms-26-00740-f002:**
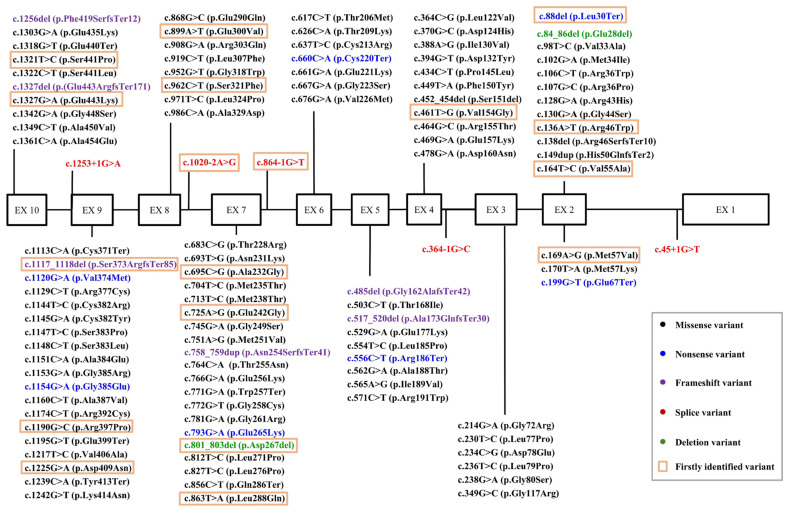
The distribution of genetic variants in the studied cohort of patients for the *GCK* gene. Genetic variants were identified in all exons except exon 1. The highest number of genetic variants was found in exons 7 and 9. No pathogenic or likely pathogenic mutations were identified in exon 1. Variants identified for the first time are highlighted with an orange frame. Missense variants are highlighted in black, nonsense variants in blue, frameshift variants in purple, splicing variants in red, and deletions in green.

**Figure 3 ijms-26-00740-f003:**
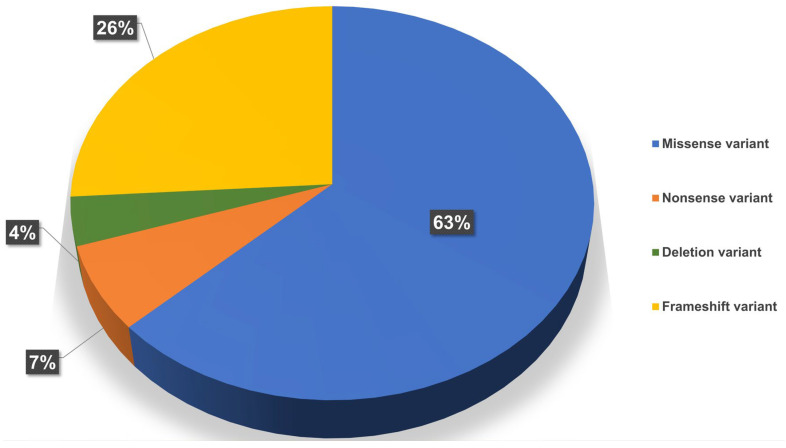
The percentage distribution of different types of genetic variants in the *HNF1A* gene in the patient cohort. Missense mutations are the most common.

**Figure 4 ijms-26-00740-f004:**
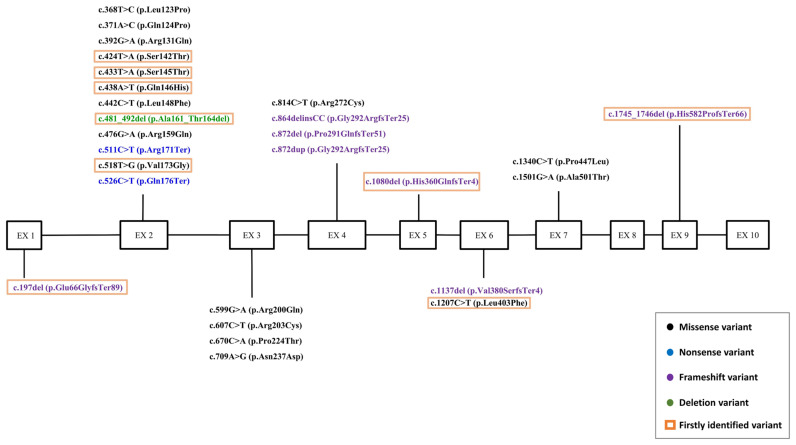
The distribution of genetic variants in the studied cohort of patients for the *HNF1A* gene. Genetic variants were identified in 8 out of 10 exons. The highest number of genetic variants was found in exon 2. Variants identified for the first time are highlighted with an orange frame. Missense variants are highlighted in black, nonsense variants in blue, frameshift variants in purple, and deletions in green.

**Table 1 ijms-26-00740-t001:** Genetic variants of the *GCK* gene identified in the study sample.

№	*GCK*	N	Mutation Type	Significance	ID
1	c.45+1G>T	2	Splice	Pathogenic	rs781260712
2	c.88del (p.Leu30Ter)	1	Nonsense	Likely pathogenic	No description
3	c.84_86del (p.Glu28del)	1	Deletion	Likely pathogenic	No ID
4	c.98T>C (p.Val33Ala)	1	Missense	Pathogenic	rs1554335954
5	c.102G>A (p.Met34Ile)	1	Missense	Likely pathogenic	rs2096283252
6	c.106C>T (p.Arg36Trp)	4	Missense	Pathogenic	rs762263694
7	c.107G>C (p.Arg36Pro)	2	Missense	Pathogenic	rs193922261
8	c.128G>A (p.Arg43His)	3	Missense	Pathogenic	rs764232985
9	c.130G>A (p.Gly44Ser)	1	Missense	Pathogenic	rs267601516
10	c.136A>T (p.Arg46Trp)	1	Missense	Likely pathogenic	No description
11	c.138del (p.Arg46SerfsTer10)	1	Deletion	Likely pathogenic	No description
12	c.149dup (p.His50GlnfsTer2)	1	Duplication	Likely pathogenic	No ID
13	c.164T>C (p.Val55Ala)	1	Missense	Likely pathogenic	No description
14	c.169A>G (p.Met57Val)	1	Missense	Likely pathogenic	No description
15	c.170T>A (p.Met57Lys)	1	Missense	Likely pathogenic	No ID
16	c.199G>T (p.Glu67Ter)	1	Nonsense	Likely pathogenic	No ID
17	c.214G>A (p.Gly72Arg)	2	Missense	Pathogenic	rs193922289
18	c.230T>C (p.Leu77Pro)	1	Missense	Likely pathogenic	rs2096281827
19	c.234C>G (p.Asp78Glu)	1	Missense	Likely pathogenic	No ID
20	c.236T>C (p.Leu79Pro)	1	Missense	Likely pathogenic	No ID
21	c.238G>A (p.Gly80Ser)	1	Missense	Pathogenic	rs1554335761
22	c.349G>C (p.Gly117Arg)	1	Missense	Likely pathogenic	No ID
23	c.364-1G>C	2	Splice	Likely pathogenic	No ID
24	c.364C>G (p.Leu122Val)	1	Missense	Likely pathogenic	rs1554335616
25	c.370G>C (p.Asp124His)	1	Missense	Likely pathogenic	rs759072800
26	c.388A>G (p.Ile130Val)	1	Missense	Likely pathogenic	No ID
27	c.394G>T (p.Asp132Tyr)	1	Missense	Likely pathogenic	No ID
28	c.434C>T (p.Pro145Leu)	2	Missense	Likely pathogenic	rs1413302700
29	c.449T>A (p.Phe150Tyr)	4	Missense	Pathogenic	rs193922297
30	c.452_454del (p.Ser151del)	3	Missense	Likely pathogenic	No ID
31	c.461T>G (p.Val154Gly)	1	Missense	Likely pathogenic	No description
32	c.464G>C (p.Arg155Thr)	1	Missense	Likely pathogenic	rs1554335573
33	c.469G>A (p.Glu157Lys)	1	Missense	Pathogenic	rs1554335570
34	c.478G>A (p.Asp160Asn)	1	Missense	Likely pathogenic	rs1554335566
35	c.485del (p.Gly162AlafsTer42)	1	Deletion	Pathogenic	rs1246464603
36	c.503C>T (p.Thr168Ile)	1	Missense	Pathogenic	No ID
37	c.517_520del (p.Ala173GlnfsTer30)	1	Deletion	Likely pathogenic	No ID
38	c.529G>A (p.Glu177Lys)	1	Missense	Likely pathogenic	No ID
39	c.554T>C (p.Leu185Pro)	1	Missense	Pathogenic	No ID
40	c.556C>T (p.Arg186Ter)	2	Nonsense	Pathogenic	rs104894006
41	c.562G>A (p.Ala188Thr)	1	Missense	Pathogenic	rs751279776
42	c.565A>G (p.Ile189Val)	1	Missense	Likely pathogenic	rs757978639
43	c.571C>T (p.Arg191Trp)	1	Missense	Pathogenic	rs1085307455
44	c.617C>T (p.Thr206Met)	7	Missense	Pathogenic	rs1441649062
45	c.626C>A (p.Thr209Lys)	1	Missense	Likely pathogenic	No ID
46	c.637T>C (p.Cys213Arg)	5	Missense	Likely pathogenic	No ID
47	c.660C>A (p.Cys220Ter)	4	Nonsense	Pathogenic	No ID
48	c.661G>A (p.Glu221Lys)	3	Missense	Pathogenic	rs193922317
49	c.667G>A (p.Gly223Ser)	5	Missense	Pathogenic	rs1360415315
50	c.676G>A (p.Val226Met)	1	Missense	Pathogenic	rs148311934
51	c.683C>G (p.Thr228Arg)	1	Missense	Pathogenic	No ID
52	c.693T>G (p.Asn231Lys)	1	Missense	Likely pathogenic	No ID
53	c.695C>G (p.Ala232Gly)	1	Missense	Likely pathogenic	No description
54	c.704T>C (p.Met235Thr)	1	Missense	Pathogenic	rs193922323
55	c.713T>C (p.Met238Thr)	1	Missense	Likely pathogenic	No ID
56	c.725A>G (p.Glu242Gly)	3	Missense	Likely pathogenic	No description
57	c.745G>A (p.Gly249Ser)	1	Missense	Likely pathogenic	rs763631453
58	c.751A>G (p.Met251Val)	1	Missense	Pathogenic	No ID
59	c.758_759dup (p.Asn254SerfsTer41)	1	Duplication	Likely pathogenic	No ID
60	c.764C>A (p.Thr255Asn)	1	Missense	Likely pathogenic	No ID
61	c.766G>A (p.Glu256Lys)	2	Missense	Pathogenic	rs769268803
62	c.771G>A (p.Trp257Ter)	1	Nonsense	Pathogenic	No ID
63	c.772G>T (p.Gly258Cys)	4	Missense	Pathogenic	rs1583596378
64	c.781G>A (p.Gly261Arg)	4	Missense	Pathogenic	rs104894008
65	c.793G>A (p.Glu265Lys)	2	Missense	Pathogenic	rs104894011
66	c.801_803del (p.Asp267del)	1	Deletion	Likely pathogenic	No description
67	c.812T>C (p.Leu271Pro)	1	Missense	Pathogenic	rs193922332
68	c.827T>C (p.Leu276Pro)	1	Missense	Pathogenic	No ID
69	c.856C>T (p.Gln286Ter)	1	Nonsense	Pathogenic	No ID
70	c.863T>A (p.Leu288Gln)	1	Missense	Likely pathogenic	No description
71	c.864-1G>T	1	Splice	Likely pathogenic	No description
72	c.868G>C (p.Glu290Gln)	1	Missense	Likely pathogenic	No ID
73	c.899A>T (p.Glu300Val)	1	Missense	Likely pathogenic	No description
74	c.908G>A (p.Arg303Gln)	1	Missense	Likely pathogenic	rs1312678560
75	c.919C>T (p.Leu307Phe)	1	Missense	Likely pathogenic	No ID
76	c.952G>T (p.Gly318Trp)	1	Missense	Pathogenic	rs193922340
77	c.962C>T (p.Ser321Phe)	1	Missense	Likely pathogenic	No description
78	c.971T>C (p.Leu324Pro)	1	Missense	Likely pathogenic	rs193922341
79	c.986C>A (p.Ala329Asp)	1	Missense	Likely pathogenic	No ID
80	c.1020-2A>G	1	Splice	Likely pathogenic	No description
81	c.1113C>A (p.Cys371Ter)	1	Nonsense	Pathogenic	rs556581174
82	c.1117_1118del (p.Ser373ArgfsTer85)	1	Deletion	Likely pathogenic	No description
83	c.1120G>A (p.Val374Met)	1	Missense	Likely pathogenic	rs1415041911
84	c.1129C>T (p.Arg377Cys)	1	Missense	Pathogenic	rs1471992838
85	c.1144T>C (p.Cys382Arg)	1	Missense	Pathogenic	rs1554334613
86	c.1145G>A (p.Cys382Tyr)	1	Missense	Pathogenic	No ID
87	c.1147T>C (p.Ser383Pro)	1	Missense	Likely pathogenic	No ID
88	c.1148C>T (p.Ser383Leu)	2	Missense	Pathogenic	rs777870079
89	c.1151C>A (p.Ala384Glu)	1	Missense	Likely pathogenic	No ID
90	c.1153G>A (p.Gly385Arg)	1	Missense	Pathogenic	rs193922267
91	c.1154G>A (p.Gly385Glu)	1	Missense	Likely pathogenic	No ID
92	c.1160C>T (p.Ala387Val)	1	Missense	Pathogenic	rs193921338
93	c.1174C>T (p.Arg392Cys)	1	Missense	Pathogenic	rs1167124132
94	c.1190G>C (p.Arg397Pro)	1	Missense	Likely pathogenic	No description
95	c.1195G>T (p.Glu399Ter)	1	Nonsense	Pathogenic	No ID
96	c.1217T>C (p.Val406Ala)	2	Missense	Likely pathogenic	No ID
97	c.1225G>A (p.Asp409Asn)	1	Missense	Likely pathogenic	No description
98	c.1239C>A (p.Tyr413Ter)	1	Nonsense	Likely pathogenic	No ID
99	c.1242G>T (p.Lys414Asn)	2	Missense	Likely pathogenic	No ID
100	c.1253+1G>A	1	Splice	Likely pathogenic	rs1394266353
101	c.1256del (p.Phe419SerfsTer12)	1	Deletion	Pathogenic	No ID
102	c.1303G>A (p.Glu435Lys)	1	Missense	Likely pathogenic	rs2096270615
103	c.1318G>T (p.Glu440Ter)	1	Nonsense	Pathogenic	No ID
104	c.1321T>C (p.Ser441Pro)	1	Missense	Likely pathogenic	No description
105	c.1322C>T (p.Ser441Leu)	1	Missense	Likely pathogenic	rs1286804191
106	c.1327del (p.(Glu443ArgfsTer171)	1	Deletion	Likely pathogenic	No description
107	c.1327G>A (p.Glu443Lys)	1	Missense	Likely pathogenic	No ID
108	c.1342G>A (p.Gly448Ser)	1	Missense	Likely pathogenic	No ID
109	c.1349C>T (p.Ala450Val)	1	Missense	Likely pathogenic	No ID
110	c.1361C>A (p.Ala454Glu)	3	Missense	Likely pathogenic	rs1057524900
111	del44144864-44189228 (44364 bp)	1	Long Deletion	Likely pathogenic	No description
Total	111	161			20

**Table 2 ijms-26-00740-t002:** Variants of the *HNF1A*, *HNF1B* and *HNF4A* genes identified in the study sample.

№		N	Mutation Type	Significance	ID
*HNF1A*					
1	c.197del (p.Glu66GlyfsTer89)	1	Deletion	Likely pathogenic	No description
2	c.368T>C (p.Leu123Pro)	1	Missense	Likely pathogenic	No ID
3	c.371A>C (p.Gln124Pro)	1	Missense	Likely pathogenic	No ID
4	c.392G>A (p.Arg131Gln)	1	Missense	Likely pathogenic	rs753998395
5	c.424T>A (p.Ser142Thr)	1	Missense	Likely pathogenic	No description
6	c.433T>A (p.Ser145Thr)	1	Missense	Likely pathogenic	No description
7	c.438A>T (p.Gln146His)	1	Missense	Likely pathogenic	No description
8	c.442C>T (p.Leu148Phe)	1	Missense	Likely pathogenic	No ID
9	c.481_492del (p.Ala161_Thr164del)	1	Deletion	Likely pathogenic	No description
10	c.476G>A (p.Arg159Gln)	1	Missense	Pathogenic	rs1172328722
11	c.511C>T (p.Arg171Ter)	1	Nonsense	Pathogenic	rs1057520291
12	c.518T>G (p.Val173Gly)	1	Missense	Likely pathogenic	No description
13	c.526C>T (p.Gln176Ter)	1	Nonsense	Pathogenic	rs754728827
14	c.599G>A (p.Arg200Gln)	1	Missense	Pathogenic	rs893256143
15	c.607C>T (p.Arg203Cys)	1	Missense	Pathogenic	rs1180119907
16	c.670C>A (p.Pro224Thr)	1	Missense	Likely pathogenic	No ID
17	c.709A>G (p.Asn237Asp)	1	Missense	Pathogenic	No ID
18	c.814C>T (p.Arg272Cys)	1	Missense	Pathogenic	rs1555212014
19	c.864delinsCC (p.Gly292ArgfsTer25)	1	Deletion-insertion	Pathogenic	rs1593058932
20	c.872del (p.Pro291GlnfsTer51)	1	Deletion	Pathogenic	rs587776825
21	c.872dup (p.Gly292ArgfsTer25)	3	Duplication	Pathogenic	rs587776825
22	c.1080del (p.His360GlnfsTer4)	1	Deletion	Likely pathogenic	No description
23	c.1137del (p.Val380SerfsTer4)	2	Deletion	Pathogenic	rs1555212248
24	c.1207C>T (p.Leu403Phe)	1	Missense	Likely pathogenic	No description
25	c.1340C>T (p.Pro447Leu)	2	Missense	Pathogenic	rs137853236
26	c.1501G>A (p.Ala501Thr)	1	Missense	Likely pathogenic	rs371807951
27	c.1745_1746del (p.His582ProfsTer66)	1	Deletion	Likely pathogenic	No description
*HNF4A*					
28	c.629C>A (p.Ala210Asp)	1	Missense	Likely pathogenic	No description
29	c.994C>T (p.Gln332Ter)	1	Nonsense	Likely pathogenic	No ID
30	del44355530-44356128 (598 п.н.)	1	Deletion	Likely pathogenic	No description
	*HNF1B*				
31	c.475C>A (p.Pro159Thr), c.491A>G (p.Lys164Arg)	1	Missense	Likely pathogenic	No description, rs1206417995
32	c.544+3_544+6del	1	Splice	Pathogenic	No ID
33	c.599A>T (p.Asp200Val)	1	Missense	Likely pathogenic	No description
Total	33	37			

**Table 3 ijms-26-00740-t003:** Genetic variants of the *ABCC8* gene identified in the study sample.

№	Variant	N	Mutation Type	Significance	ID
1	c.307C>T (p.His103Tyr)	1	Missense	Likely pathogenic	rs751209734
2	c.742C>T (p.Arg248Ter)	1	Nonsense	Pathogenic	rs72559730
3	c.1303T>C (p.Cys435Arg)	1	Missense	Likely pathogenic	No ID
4	c.1562G>A (p.Arg521Gln)	1	Missense	Likely pathogenic	rs368114790
5	c.1811T>C (p.Leu604Pro)	1	Missense	Likely pathogenic	No description
6	c.2389C>T (p.Arg797Trp)	1	Missense	Likely pathogenic	rs142620721
7	c.2473C>T (p.Arg825Trp)	1	Missense	Pathogenic	rs779736828
8	c.2475+5G>T	1	Splice	Likely pathogenic	rs778765864
9	c.2473C>T (p.Arg825Trp)	1	Missense	Pathogenic	rs779736828
10	c.2488T>A (p.Ser830Thr)	1	Missense	Likely pathogenic	No ID
11	c.2798G>A (p.Arg933Gln)	1	Missense	Likely pathogenic	rs745591375
12	c.3110C>T (p.Thr1037Ile)	1	Missense	Likely pathogenic	No description
13	c.3443T>C (p.Leu1148Pro)	1	Missense	Likely pathogenic	No description
14	c.3748C>T (p.Arg1250Ter)	1	Nonsense	Pathogenic	rs1057516281
15	c.3989-9G>A	1	Splice	Pathogenic	rs151344623
16	c.4454A>C (p.Gln1485Pro)	1	Missense	Likely pathogenic	No description
17	c.4516G>A (p.Glu1506Lys)	1	Missense	Pathogenic	rs137852671

**Table 4 ijms-26-00740-t004:** Genetic variants of the *INS* and *INSR* genes identified in the study sample.

№	Vaiant	N	Mutation Type	Significance	ID
	*INS*				
1	c.163C>T (p. Arg55Cys)	1	Missense	Pathogenic	rs121908261
2	c.94G>A (p.Gly32Ser)	1	Missense	Pathogenic	rs80356664
3	c.29T>A (p.Leu10Gln)	1	Missense	Likely pathogenic	No description
4	c.265C>T (p.Arg89Cys)	1	Missense	Pathogenic	rs80356669
	*INSR*				
1	c.322G>T (p.Val108Phe)	1	Missense	Likely pathogenic	No description
2	c.1573C>T (p.Arg525Ter)	1	Missense	Likely pathogenic	rs1599937180
3	c.1622A>G (p.Asn541Ser)	1	Missense	Likely pathogenic	No description
4	c.2621C>T (p.Pro874Leu)	1	Missense	Llikely pathogenic	No ID
5	c.3659+1G>C	1	Splice	Likely pathogenic	No description

**Table 5 ijms-26-00740-t005:** Other genetic variants identified in the study sample.

№	Variant	N	Mutation Type	Pathogenic	ID
				*KCNJ11*	
1	c.526C>T (p.Arg176Cys)	1	Missense	Likely pathogenic	rs201264306
2	c.602G>A (p.Arg201His)	1	Missense	Pathogenic	rs80356624
3	c.149G>A (p.Arg50Gln)	2	Missense	Pathogenic	rs80356611
				*PAX4*	
4	c.638C>T (p.Thr213Met)	1	Missense	Likely pathogenic	rs528075802
5	c.771+3A>G	1	Splice	Likely pathogenic	rs776955589
				*PDX1*	
6	c.417C>G (p.Tyr139Ter)	1	Missense	Likely pathogenic	No description
7	c.533A>C (p.Glu178Ala)	1	Missense	Likely pathogenic	No description
				*ZFP57*	
8	c.115del (p.Val39Ter)	1	Nonsense	Likely pathogenic	rs758356804
				*BLK*	
9	c.590C>A (p.Ser197Ter)	1	Nonsense	Likely pathogenic	rs150630845
				*WFS1*	
10	c.1214A>G (p.Tyr405Cys)	1	Missense	Likely pathogenic	rs769514478

**Table 6 ijms-26-00740-t006:** Sample characteristics.

Guiding Diagnosis	Median Age (μ ± Sd)	Quantity (N)
Diabetes mellitus, unspecified	34.50 ± 10.50	43
MODY without concomitant pathologies	13.00 ± 6.89	337
Suspected monogenic forms of diabetes	14.00 ± 9.51	68
Diabetes with associated complications	14.00 ± 3.00	15
Congenital hyperinsulinism	1.00 ± 1.68	16
Neonatal diabetes mellitus	1.00 ± 4.84	27

## Data Availability

The data from this study can be obtained from the corresponding author upon making a reasonable request if there are no privacy or ethical issues.

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
