# Peer review of "Genetic Structure of Hereditary Forms of Diabetes Mellitus in Russia"

_ijms, 2025, doi:10.3390/ijms26020740_

Round 1
Reviewer 1 Report
Comments and Suggestions for Authors
The research is of interest because it addresses a growing current topic: precision medicine in diabetes. To achieve this, it is necessary to optimize diagnostic and preventive algorithms, which cannot be done without a thorough analysis of the genetic architecture of hereditary forms of diabetes in different populations with diverse ethnic characteristics. The more we understand the various genetic mutations associated with diabetes, the better the chances of providing precise responses to diagnoses and/or treatments for the disease in different population subgroups with similar characteristics, thus minimizing error and risk, while maximizing the effectiveness of decisions regarding disease management. Therefore, the article reviewed seems particularly relevant, especially since it addresses a population that has been under-studied, such as the Russian population.
The research presented identifies a high number of pathogenic or possibly pathogenic variants (180) in 13 genes (GCK, HNF1A, HNF1B, HNF4A, ABCC8, INS, INSR, KCNJ11, PAX4, PDX1, ZFP57, BLK, WFS1), identified in about half of the population studied. Of all the genes, the glucokinase gene (GCK) presents the highest number of identified variants, some of which are identified for the first time. The same is true for the tissue-specific transcription factor genes HNF1A, HNF4A, and HNF1B, where some previously unreported variants were also detected. They also report other variants found in the ABCC8 gene, the insulin gene (INS), and its receptor gene (INSR). Thus, they identify these genes (GCK, HNF1A, HNF1B, HNF4A, ABCC8, INS, and INSR) as the primary genetic factors to consider in the pathogenic development of hereditary forms of diabetes mellitus in Russia.
Therefore, the objective of the publication—to find pathogenic variants in specific genes in Russian patients with suspected hereditary forms of diabetes—is perfectly clear and theoretically supported in the introduction, and achieved in the results and discussion.
The description of the methodology is correct. The different methods used are well described, as well as the inclusion methods for participants, with the study sample being acceptable.
The presentation of the results seemed adequate to me, both in the text and in the images provided, which allow for a more direct visualization of the genetic variants identified in the study sample.
Regarding the discussion, I also found it correct, though somewhat brief. Perhaps the results could have been compared with more population groups. This observation is also supported by the fact that the bibliographic references are somewhat scarce, totaling 28. However, it is worth highlighting that this bibliography is quite up to date, as 68% of the citations are from works published between 2020 and 2024. However, the population variety is limited.
Based on the above, I consider the work suitable for publication.

Author Response
Comment 1:
"Regarding the discussion, I also found it correct, though somewhat brief. Perhaps the results could have been compared with more population groups. This observation is also supported by the fact that the bibliographic references are somewhat scarce, totaling 28. However, it is worth highlighting that this bibliography is quite up to date, as 68% of the citations are from works published between 2020 and 2024. However, the population variety is limited."
Response:
Thank you for your valuable feedback. We appreciate your suggestion to expand the discussion and provide comparisons with additional population groups. In response, we have included comparisons with studies conducted in sub-Saharan Africa, Korea, the Middle East, and other regions. These additions highlight the similarities and differences in the genetic architecture of MODY across diverse populations. Additionally, we have expanded the bibliographic references to include these studies, increasing the total number of references from 28 to 34. This revision provides a broader perspective on the genetic diversity of MODY and enriches the discussion.
Changes made:
We have added the following text to the discussion section on pages 19-21:
- «Our findings on the prevalence and spectrum of MODY-associated mutations in the Russian population align with global trends, while also highlighting regional genetic characteristics. A comprehensive analysis of MODY in sub-Saharan Africa revealed significant heterogeneity in MODY-related gene variants, with novel mutations identified in GCK, HNF1A, and ABCC8 genes. Interestingly, the frequency of GCK mutations in African cohorts was considerably lower compared to European and Russian populations, suggesting potential regional genetic influences on MODY subtype distribution. These findings underscore the importance of exploring population-specific genetic landscapes to improve diagnostic accuracy in diverse ethnic groups» [Behl, R., Malhotra, N., Joshi, V. et al. Meta-analysis of HNF1A-MODY3 variants among human population. J Diabetes Metab Disord 21, 1037–1046 (2022). https://doi.org/10.1007/s40200-022-00975-8].
- «Studies conducted in Korean populations demonstrated a high prevalence of HNF1A and HNF4A mutations in individuals with early-onset diabetes, mirroring our findings in the Russian cohort. However, unique mutations were identified in mitochondrial DNA, which are rare in European and Russian populations. This observation highlights the potential for distinct genetic contributions in Asian populations and the necessity of including mitochondrial gene analysis in diagnostic protocols for MODY» [Dong-Hwa Lee, Soo-Heon Kwak, Hee Sue Park, Eu Jeong Ku, Hyun Jeong Jeon, Tae Keun Oh - Identification of candidate gene variants of monogenic diabetes using targeted panel sequencing in early onset diabetes patients: BMJ Open Diabetes Research & Care 2021; 9:e002217].
- A systematic review of MODY mutations worldwide reported that GCK and HNF1A mutations remain the most prevalent across all populations studied, comprising up to 80% of cases. However, the same review emphasized the underrepresentation of data from Eastern Europe, including Russia, which our study helps to address. By identifying novel mutations in MODY-related genes within the Russian cohort, we contribute to filling this gap and expanding the global understanding of MODY's genetic spectrum [Rafique, I., Mir, A., Saqib, M.A.N. et al. Causal variants in Maturity Onset Diabetes of the Young (MODY) – A systematic review. BMC Endocr Disord 21, 223 (2021). https://doi.org/10.1186/s12902-021-00891-7].
- Another notable comparison can be drawn with Middle Eastern populations, where the frequency of mutations in the INS and HNF1B genes is higher compared to other regions. While these mutations are rare in our cohort, the novel variants identified in the Russian population underscore the distinct genetic environment influenced by ethnic and geographical factors. These disparities highlight the necessity for tailored genetic studies to address region-specific clinical needs [Sara Asgarian, Hossein Lanjanian, Shiva Rahimipour Anaraki et al. From Genes to Diagnosis: Examining the Clinical and Genetic Spectrum of Maturity-Onset Diabetes of the Young (MODY) in TCGS, 22 February 2024, PREPRINT (Version 1) available at Research Square [https://doi.org/10.21203/rs.3.rs-3927463/v1]].
- Recent reviews on MODY genetics have noted the significant overlap in clinical presentations between MODY subtypes and other forms of diabetes, complicating diagnosis. Our study's use of next-generation sequencing not only validates previously reported global findings but also introduces novel insights into genotype-phenotype correlations specific to the Russian population. These findings emphasize the role of high-throughput sequencing in uncovering unique genetic variants that contribute to diabetes pathogenesis in underrepresented populations [Hasballa, I.; Maggi, D. MODY Only Monogenic? A Narrative Review of the Novel Rare and Low-Penetrant Variants. J. Mol. Sci. 2024, 25, 8790. https://doi.org/10.3390/ijms25168790].
Reviewer 2 Report
Comments and Suggestions for Authors
The study demonstrates the gene variations in patients of diabetes in Russia. Variants in the genes GCK, HNF1A, HNF1B, HNF4A, ABCC8, INS, and INSR may be responsible for pathogenicity of diabetes in Russia.
1. Section 4. Materials and Methods need to be revised to describe the procedure of obtaining written informed consent and IRB protocol numbers, etc.
2. Conclusion needs to be elaborated for describing the importance of the gene variations identified in the study.
3. Figures need precise legends to indicate the types of variants more in detail.
Author Response
Comment 1:
"Section 4. Materials and Methods need to be revised to describe the procedure of obtaining written informed consent and IRB protocol numbers, etc."
Response:
Thank you. We revised Section 4 to include a detailed description of the ethical approval process, the IRB protocol numbers, and the procedure for obtaining written informed consent from all participants.
Changes made:
The following text was added to Section 4 on page 21:
"Ethical approval for this study was obtained from the Institutional Review Board (IRB) of Endocrinology research center under protocol number 16 dated 13 September 2023. Written informed consent was obtained from all participants or their legal guardians in accordance with the Declaration of Helsinki. The consent form detailed the purpose of the study, procedures involved, potential risks, and the confidentiality of personal data."
Comment 2:
"Conclusion needs to be elaborated for describing the importance of the gene variations identified in the study."
Response:
We expanded the Conclusion section to emphasize the clinical and scientific significance of the gene variations identified in this study.
Changes made:
The following sentences were added to the Conclusion on page 21:
"The identification of pathogenic variants in genes such as GCK, HNF1A, and ABCC8 highlights their significance in MODY’s molecular etiology within the Russian population. These findings provide a valuable basis for refining diagnostic algorithms, facilitating earlier detection, and advancing personalized treatment strategies. Moreover, the novel variants identified in this study enhance the understanding of MODY’s genetic landscape on a global scale and serve as a foundation for further studies on genotype-phenotype correlations in underrepresented populations.
Comment 3:
"Figures need precise legends to indicate the types of variants more in detail."
Response:
We have revised the figure legends to provide more detailed information about the types of variants.
Reviewer 3 Report
Comments and Suggestions for Authors
· In abstract, please state the age of patients and connect the findings to future genetic implications.
· Avoid short paragraphs in introduction-develop further or combine.
· The paper overall should be qualified with more supporting evidence (only 28 references were used). The introduction and discussion would be considerably strengthened by discussing the following: (1) classification of NDM; (2) identification of NDM susceptibility genes; (3) epigenetic alterations in NDM; and (4) current discoveries and management of NDM. Also, epigenome-wide association studies have shown that GDM can significantly affect NDM gene expression profiles in neonates (e.g., Int J Mol Sci. 2021 Aug 31;22(17):9462; Front Endocrinol (Lausanne). 2020 Dec 1:11:602477; Genes (Basel). 2023 Apr 29;14(5):101). Thus, there is a need to explore the differential expression and association of genes in both GDM and NDM.
· Authors should revise the last paragraph in introduction to better articulate the specific novel contribution of the research. Why this research is important? What is new?
· Patients recruitment should be described in much more details (Line 536-539). Please also describe data collection procedure in details.
· It is unclear how authors interpret sequence variants? Were only pathogenic or likely pathogenic sequence variants identified? What about other categories (e.g., VUS, benign, likely benign).
· The discussion lacks cohesion and many results are not sufficiently discussed. It is unclear what was found in the study and other studies that authors compared with. I suggest the authors to link the literature with the important results of this study. The limitations of the study were also not reported.
· Also need to draw conclusions with future genetic implications.
Comments on the Quality of English LanguageEnglish language needs to be improved throughout.
Author Response
Comments 1:
«In abstract, please state the age of patients and connect the findings to future genetic implications»
Response:
Thank you. Information about the average age of patients was added to the section 4 (Materials and Methods) – 13.5 years (±10.15).
Comments 2:
«Avoid short paragraphs in introduction-develop further or combine»
Response:
We combined the paragraphs in the introduction.
Comments 3:
«The paper overall should be qualified with more supporting evidence (only 28 references were used). The introduction and discussion would be considerably strengthened by discussing the following: (1) classification of NDM; (2) identification of NDM susceptibility genes; (3) epigenetic alterations in NDM; and (4) current discoveries and management of NDM. Also, epigenome-wide association studies have shown that GDM can significantly affect NDM gene expression profiles in neonates (e.g., Int J Mol Sci. 2021 Aug 31;22(17):9462; Front Endocrinol (Lausanne). 2020 Dec 1:11:602477; Genes (Basel). 2023 Apr 29;14(5):101). Thus, there is a need to explore the differential expression and association of genes in both GDM and NDM»
Response:
In the introduction, we outlined the concept and challenges associated with studying the genetics of hereditary diabetes, highlighting key scientific trends and unresolved issues in this field. We systematically described the classification and genetic factors of hereditary diabetes mellitus to maintain clarity and focus in the introduction. A more detailed analysis of the literature and a comparison with our results are presented in the discussion section. Additional information about the classification of neonatal diabetes mellitus (NDM), its genetic causes, and recent references on the genetic structure of NDM has been incorporated. The epigenetic aspects of maturity-onset diabetes of the young (MODY) and neonatal diabetes are still underexplored, with existing data primarily focused on adult patients or gestational diabetes. In this article, we have focused on the genetic aspects of diabetes.
Changes made:
Neonatal diabetes is categorized into transient and permanent forms. NDM, which manifests within the first six months of life, is categorized as a genetic form of diabetes. It occurs in approximately 1 in 350,000 births. Approximately half of all NDM cases are classified as transient. Both dominant and recessive inheritance patterns have been identified, as well as sporadic cases unrelated to inheritance [3]. The most frequent genetic causes of NDM are abnormalities of the 6q24 locus and pathogenic variants in KATP channel genes. Additionally, syndromic forms of neonatal diabetes mellitus, such as Rogers syndrome (TRMA), Wolfram syndrome, Alstrom syndrome, and Wolcott-Rallison syndrome, are characterized by genetic mutations that lead to chronic health issues. Detailed data on the genetic makeup of neonatal diabetes is comprehensively described in articles by Beltran et al. (2020) [8] and Golshan-Tafti et al. (2024) [9].
Comments 4:
«Authors should revise the last paragraph in introduction to better articulate the specific novel contribution of the research. Why this research is important? What is new?»
Response:
We revised Section 1 and expanded the last paragraph.
Changes made:
Thus, the identification of the molecular pathogenesis of hereditary forms of diabetes is far from complete, and further studies on its genetic architecture, taking into account the population-specific characteristics of patient cohorts, are crucial for advancing research. Understanding the molecular and genetic basis of diabetes in the current state of research is critical to the development of new therapeutic approaches aimed at effective disease management and the provision of accurate medical and genetic counseling to affected families.
The presented study contributes to the expansion of knowledge in the field of the molecular pathogenesis of hereditary forms of diabetes, provides insights into the genetic landscape of the disease in Russian patients, and offers valuable information on previously unreported variants, which subsequently enriches the database of pathogenic and likely pathogenic variants in target genes associated with the disease.
Comments 5:
«Patients recruitment should be described in much more details (Line 536-539). Please also describe data collection procedure in details»
Response:
We expanded patient’s recruitment and data collection procedure.
Changes made:
The study involved 506 unrelated patients treated at the Endocrinology Research Center, including 240 males (median age: 13 years) and 266 females (median age: 14 years). These patients were suspected of having hereditary forms of diabetes without signs of an autoimmune process, as indicated by the absence of detectable levels of IAA, ICA, GADA, and Zn8 antibodies at disease onset. The inclusion criteria were preserved C-peptide secretion, low demand for exogenous insulin within five years of disease onset, and the absence of severe obesity. Additional cohort characteristics, categorized by the referring physician's presumptive diagnosis, are presented in Table 6.
Data collection followed an established protocol for this patient category, covering detailed medical histories (including hereditary burden, pregnancy, and delivery history) and measurements of anthropometric and laboratory parameters. Laboratory tests included venous plasma glucose (measured in isolation or during an oral glucose tolerance test [OGTT]), glycated hemoglobin (HbA1c), C-peptide, and insulin. Patients meeting the inclusion criteria provided written informed consent and underwent venous blood sampling into K3/2EDTA tubes. Informed consent was obtained from adult patients or a parent/legal guardian for pediatric participants.
Comments 6:
«It is unclear how authors interpret sequence variants? Were only pathogenic or likely pathogenic sequence variants identified? What about other categories (e.g., VUS, benign, likely benign).»
Response:
The interpretation of DNA sequence variants was performed in accordance with the guidelines of the American College of Medical Genetics and Genomics (ACMG) for variant pathogenicity assessment (S. Richards et al., “Standards and guidelines for the interpretation of sequence variants: A joint consensus recommendation of the American College of Medical Genetics and Genomics and the Association for Molecular Pathology,” Genet. Med., vol. 17, no. 5, pp. 405–424, 2015, doi: 10.1038/gim.2015.30). The aim of this study was to present the structure of causal gene variants in monogenic forms of diabetes among patients from Russia. Therefore, variants of uncertain or benign significance were not included. More than 40 detected variants have not been previously described in the literature. Their functional significance remains to be determined through further studies, which will be presented in future publications. In this work, we focused solely on the presented results without extending the scope to include benign variants
Comments 7:
«The discussion lacks cohesion and many results are not sufficiently discussed. It is unclear what was found in the study and other studies that authors compared with. I suggest the authors to link the literature with the important results of this study. The limitations of the study were also not reported»
Response:
We have significantly expanded both the comparison of our results with the literature data and the conclusion of the article. In addition, we have added research limitations.
Changes made:
«The limitations of this study include the lack of proven pathogenic significance for some variants and the limited availability of information. Additional limitations include the absence of clinical-genetic correlation analysis, largely due to the high diversity of nucleotide sequence variations and the limited detail of information about the ethnic origin of patients.»
Comments 8:
«Also need to draw conclusions with future genetic implications»
Response:
In the conclusion of the article, we revised the final section to state that treatment adjustments will be made based on molecular genetic testing data. The prognosis of disease progression will also be determined. Furthermore, carriers of identified variants among family members will undergo comprehensive clinical and laboratory testing. Additionally, the study's limitations were outlined to provide a balanced interpretation of the findings.
Changes made:
«The identification of pathogenic variants in genes such as GCK, HNF1A, and ABCC8 highlights their significance in MODY’s molecular etiology within the Russian population. These findings provide a valuable basis for refining diagnostic algorithms, facilitating earlier detection, and advancing personalized treatment strategies. Moreover, the novel variants identified in this study enhance the understanding of MODY’s genetic landscape on a global scale and serve as a foundation for further studies on genotype-phenotype correlations in underrepresented populations.
In conclusion, our study has expanded the understanding of the molecular genetic basis of hereditary diabetes in Russia. Based on these findings, genetic diagnostic algorithms for the disease are being refined. Our results contribute to the expansion of the database of causative variants for hereditary forms of diabetes, laying a strong foundation for future functional studies.
The limitations of this study include the lack of proven pathogenic significance for some variants and the limited availability of information. Additional limitations include the absence of clinical-genetic correlation analysis, largely due to the high diversity of nucleotide sequence variations and the limited detail of information about the ethnic origin of patients.»
Comments 9:
«English language needs to be improved throughout.»
Response:
Thank you for the comment. We have tried to improve English throughout the text.
Round 2
Reviewer 2 Report
Comments and Suggestions for Authors
The authors addressed the reviewer's comments. The manuscript has been improved.
Author Response
Dear Reviewer,
We sincerely thank you for your thoughtful review of our manuscript and your positive feedback. We are pleased to hear that the revisions we made have improved the quality of the manuscript and met your expectations.
Your comments and suggestions have greatly contributed to clarifying and strengthening our work, and we truly appreciate your valuable input.
Reviewer 3 Report
Comments and Suggestions for Authors
Dear Authors,
Please connect the findings to future genetic implications in abstract. The statement in Line 29-31 could be improved.
What are current discoveries and management of NDM? Please also make a clear connection between GDM and NDM in introduction. Epigenome-wide association studies have shown that GDM can significantly affect NDM gene expression profiles in neonates (e.g., Genes (Basel). 2023 Apr 29;14(5):101; Int J Mol Sci. 2021 Aug 31;22(17):9462; Front Endocrinol (Lausanne). 2020 Dec 1:11:602477).
Author Response
Comment 1:
“Please connect the findings to future genetic implications in abstract.”
Response:
We have revised the abstract to connect the findings to potential genetic implications.
Changes made:
The following sentence was added to the abstract:
“This knowledge provides a foundation for developing region-specific genetic testing algorithms and personalized therapeutic strategies, which are critical for future precision medicine initiatives.”
Comment 2:
“The statement in Line 29-31 could be improved.”
Response:
We have rephrased the statement for better clarity and precision.
Changes made:
The original sentence was replaced with:
“Variants in the genes GCK, HNF1A, HNF1B, HNF4A, ABCC8, INS, and INSR were the main contributors to the genetic pathogenesis of hereditary diabetes mellitus in the Russian cohort. These findings enhance our understanding of the molecular mechanisms underlying the disease and provide a solid basis for future studies aimed at improving diagnostic accuracy and advancing personalized therapeutic strategies.”
Comment 3:
“What are current discoveries and management of NDM? Please also make a clear connection between GDM and NDM in the introduction. Epigenome-wide association studies have shown that GDM can significantly affect NDM gene expression profiles in neonates.”
Response:
We have expanded the introduction to include a discussion of current discoveries and management strategies for neonatal diabetes mellitus (NDM). Additionally, we incorporated references and detailed the connection between gestational diabetes mellitus (GDM) and NDM.
Changes made:
The following text was added to the introduction:
“Neonatal diabetes mellitus (NDM) is a rare genetic form of diabetes that manifests within the first six months of life, with an incidence of approximately 1 in 350,000 live births worldwide. Clinically, NDM is divided into two forms: transient and permanent. Transient NDM, accounting for about 50% of all cases, is characterized by spontaneous remission after infancy, although relapses can occur later in life. In contrast, permanent NDM persists throughout life.
The causes of NDM encompass hereditary as well as sporadic cases. The most common genetic causes are abnormalities in the 6q24 locus and pathogenic mutations in genes encoding KATP channels, which are critical regulators of insulin secretion. In addition, epigenetic studies have shown that gestational diabetes mellitus (GDM) can significantly influence neonatal gene expression profiles. According to recent epigenome-wide studies, GDM can alter the methylation of genes involved in insulin regulation and pancreatic development and is associated with an increased risk of NDM in offspring [35-36].
Syndromic forms of NDM, such as Rogers syndrome (TRMA), Wolfram syndrome, Alstrom syndrome, and Wolcott-Rallison syndrome, serve as examples of the genetic heterogeneity of the disease. These forms are accompanied by multiple chronic comorbidities resulting from mutations disrupting the function of several physiological systems. A detailed analysis of the genetic landscape of NDM has been provided in studies by Beltran et al. (2020) [8] and Golshan-Tafti et al. (2024) [9], which highlight the molecular diversity underlying this condition.
Advances in the management of NDM have been made possible by the implementation of high-throughput sequencing technologies, which enable accurate genetic characterization. This facilitates targeted therapies, such as the use of sulfonylureas for patients with KATP channelopathies. Additionally, environmental factors during pregnancy, particularly maternal glycemic control, significantly influence gene expression and disease progression in children. Thus, managing NDM requires an integrative approach that combines personalized therapies targeting genetic and epigenetic factors with consideration of environmental influences. These strategies open new opportunities to improve the diagnosis, treatment, and long-term outcomes of patients with NDM.”